# Melanotrichoblastoma: A Histopathological Case Report of a Rare Pigmented Variant of Trichoblastoma

**DOI:** 10.3390/reports8030130

**Published:** 2025-08-01

**Authors:** George S. Stoyanov, Ivaylo Balabanov, Svetoslava Zhivkova, Hristo Popov

**Affiliations:** 1Department of Pathology, Multiprofile Hospital for Active Treatment, 9700 Shumen, Bulgaria; 2Department of Obstetrics and Gynecology, Multiprofile Hospital for Active Treatment, 9700 Shumen, Bulgaria; 3Department of General and Clinical Pathology, Forensic Medicine and Deontology, Faculty of Medicine, Medical University—Varna, 9002 Varna, Bulgaria

**Keywords:** melanotrichoblastoma, trichoblastoma, pigmented tumor, melanocytes, skin tumor, appendiceal differentiation

## Abstract

**Background and clinical significance:** Trichoblastomas are rare, mixed epithelial tumors with a mesenchymal component and hair follicle differentiation. **Case presentation:** Herein, we present a case report of a 51-year-old female patient presenting to the obstetrics and gynecology department with complaints of edema and erythema of the right Bartholin gland, and a lesion measuring 2 cm on the right lateral edge of the mons pubis, towards the inguinal fold. Marsupialization of the Bartholin gland was performed, as well as an incision into the pubo-inguinal lesion, which the patient depicted as grossly resembling an ingrown hair. Upon incision into the pubic–inguinal lesion, it was dark brown in color and spontaneously popped out of the subcutis, without an attempt at enucleation. Histology and subsequent immunohistochemistry of the lesion showed a blue basaloid tumor with an extensive pigment component located deep in the dermis that was sharply demarcated from the surrounding tissues. **Conclusion:** Immunohistochemistry was diffusely and strongly positive for epithelial markers; melanocytic markers were positive only in dendritic melanocytes dispersed within the tumors, and the proliferative index was low. As such, the tumor was identified as melanotrichoblastoma.

## 1. Introduction and Clinical Significance

Trichoblastomas are rare, benign tumors of the skin with hair follicle differentiation [1]. The nosological unit of these rare neoplasms encompasses several distinct entries such as conventional trichoblastoma, columnar or desmoplastic trichoblastoma, trichoepithelioma, and trichogerminomas [1,2]. While generally considered epithelial tumors, they are a great example of a mixed tumor as they generally reenact the embryonic development of hair follicles with distinct epithelial and mesenchymal components [1].

These neoplasms are generally rare, and their true incidence in the population is unknown [1]. Their predilection sites are the head and neck area, as well as the pelvic girdle [1]. Tumors are most often diagnosed in the fifth and sixth decades of life, with some genetic syndromes, such as Brooke–Spiegler syndrome and multiple familial trichoepithelioma, leading to increased incidence of these tumors as well as an earlier age of diagnosis [1].

Trichoblastomas were first described by Brooke, for whom the aforementioned syndrome is named as epithelioma adenoides cysticum [1]. For a long period of time, these neoplasms were seen as a rare and indolent subtype of basal cell carcinoma based on the age of the presenting patient, their location, and significant morphological similarities with them. Over a long period of time, many contributions were made to the definition of these tumors by authors such as Pinkus, Lever and Montgomery, and Headington and Ackerman, both to distinguish them from basal cell carcinoma morphologically and to define their origin from hair follicle germinative cells [1].

Trichoblastomas are probably underrecognized tumors, as their age and location of incidence, as well as morphology of a dermal basaloid tumor, often lead to them being misclassified as basal cell carcinomas, with only a few morphological features distinguishing them from the aforementioned malignancy [1,2,3]. The key differential criteria are that trichoblastomas are located deep into the reticular dermis, while basal cell carcinoma, even if extending to the reticular dermis and subcutis, always has an epidermal or hair shaft connection, which trichoblastomas lack [2,3]. While both tumors are basaloid and often grow in a nest-like pattern with peripheral cellular palisading, basal cell carcinomas often have clefting towards the surrounding dermis due to mucin production, while trichoblastomas encompass a specific mesenchymal stroma distinct from the dermis, with formation of papillary mesenchymal bodies towards the nest [2]. Last but not least, as trichoblastomas recapitulate the structure of the embryonal hair follicle, they often have a distinct third component comprising interepithelial Merkel cells, which could be distinguished via cytokeratin 20 (CK20), with basal cell carcinoma almost always lacking such a component [1,2].

An even rarer occurrence, further confirming the embryonic hair follicle recapitulation of these tumors, is the presence of melanocytes within the tumors. Such entries have only been described within the last two decades and have been designated as melanotrichoblastoma [4]. Due to the rarity of these tumors, the official World Health Organization (WHO) classification of skin tumors has not yet included them as a separate entry or distinct type of trichoblastoma, as to date, only six such cases have been published. As the presence of melanocytes and melanin pigment in these entries is often striking, these tumors must always be distinguished further from pigmented basal cell carcinoma, a recognized WHO subtype, and melanoma.

Herein, we report a case of melanotrichoblastoma with pelvic girdle localization.

## 2. Case Presentation

A previously healthy 51-year-old female presented to the Department of Obstetrics and Gynecology with complaints of edema and erythema in the topographical projection of the right Bartholin gland, accompanied by discomfort for the previous month. During clinical evaluation, a lesion measuring 2 cm was palpated on the right lateral edge of the mons pubis in the projection of the inguinal ligament, which grossly was deeply located and non-adherent to the neighboring tissues, resembling an ingrown hair. The patient had no recollection of the timeframe of origin of the lesion, but stated that it had been present for some time.

Due to the clinical symptoms, a marsupialization of the Bartholin gland was performed, with incision into the pubo-inguinal lesion to differentiate an ingrown hair from a reactively enlarged lymph node. Upon excision, the lesion was dark brown in color and spontaneously popped out of the subcutis, without an attempt at enucleation. Due to this finding and under suspicion of a tumor with extensive necrosis, the neighboring tissues and skin were excised completely.

Grossly, the excited specimen was represented by normal cutis and subcutis and fleshy brown tissue with a nodular appearance. Histopathology showed intact epidermis and dermis with a deeply located, basaloid nested tumor within the reticular dermis, with extensive brown pigment deposition (Figure 1). The tumor itself was located deep in the dermis with subcutaneous extension and was sharply demarcated from the surrounding tissues, with a focal appearance of an intracystic tumor due to significant detachments from them. Growth pattern was represented by nests of various calibers and reticular-papillary structures, with some of the nests exhibiting a basaloid appearance and focal peripheral palisading with a complete absence of epidermal and cutaneous–adnexal connection. Some of the nests formed primitive hair follicular rudiments with mesenchymal papillary bodies and pronounced basaloid type morphology. Larger tumor cell nests had foci of interstitial edema with the formation of cystic spaces of various calibers, focally filled with eosinophilic matter, including areas similar to primitive hair structures. In these nests, the cells were large with central bright nuclei and marginated chromatin, without clearly visible boundaries between the cells. Focally degenerative type atypia was noted along with sparse mitoses in the entire tumor population. Some areas showed an onion-skin-like cellular arrangement of paler cells. The tumor itself was extensively colonized by melanocytes with pronounced melanin production and incontinence with accumulation of pigment in a small part of the peripheral tumor cells, pigmented macrophages, and freely in the intercellular space.

The tumor was interpreted as a large nodular and retiform trichoblastoma of the pigmented type. Follow-up immunohistochemistry was performed with pan-cytokeratin (CK AE1/AE3) and epithelial cell adhesion molecule (BerEp4) to prove the epithelial component, melan-A and human melanoma black 45 (HMB-45) to prove the melanocyte component, and proliferative index (Ki-67) to underline the benign nature of the tumor.

Both CK AE1/AE3 and BerEP4 were diffusely positive, with the typical location of the reaction—cytoplasmic for CK AE1/AE3 and membranous for BerEP4 (Figure 2). Melan-A and HMB-45 were also positive, both in a typical cytoplasmic manner, only in the melanocytic tumor cell component, and underlining the hair-follicle-like spindle and dendritic cell morphology of the cells (Figure 3). The Ki-67 proliferative index was high overall at 30%, with most of the positive cells being located centrally within the tumor nests, while peripheral cells had a very low proliferative index (Figure 2). As such, the tumor was confirmed via immunohistochemistry to be a melanotrichoblastoma.

Despite melanotrichoblastomas being benign tumors, due to their exceedingly rare nature and the location of the tumor, the patient is regularly followed up in an outpatient setting, with no local recurrence or distant spread being noted on the eight-month point of follow-up.

## 3. Discussion

Melanotrichoblastomas are exceedingly rare and probably underreported tumors. To date, there have been six published cases of melanotrichoblastoma, with ours being the seventh overall. In a seminal work on the topic, published in 2023, Ocanha-Xavier and Xavier-Júnior reported the sixth published case and reviewed the previous five [5]. Based on the literature review part of their manuscript, the currently reported case is the first case of melanotrichoblastoma reported on the pelvic girdle, with most of the previously reported cases being located in the head and neck region. Age of diagnosis varies from 25 to 72 years of age, with a female predominance.

The first reported case, which also coined the terminology of this exceedingly rare tumor, was published in 2002 by Kanitakis et al., with the following cases being published in 2011 by Kim et al., 2012 by Hung et al., 2017 by Quist and DiMaio, 2021 by Mizuta et al., and the aforementioned case in 2023 by Ocanha-Xavier and Xavier-Júnior [4,5,6,7,8,9]. Interestingly, while conventional trichoblastoma is often associated with a sebaceous nevus (Jadassohn nevus), only the case published by Hung et al. in 2012 has shown such an association for melanotrichoblastoma as one newer reported case [9,10]. As seen in our case, there was no association with a sebaceous nevus as well.

As already mentioned, the main differential diagnosis of trichoblastoma is that of basal cell carcinoma, which carries over to melanotrichoblastoma as pigmented basal cell carcinoma can also have an extensive pigment deposition due to hemosiderin aggregation as well as dispersed spindle melanocytes [1,2,3]. Although deceivingly easy at first glance, the differential diagnosis between the two can be difficult, as basal cell carcinoma can have an association (origin) not only from the epidermal lining, but also with the cutaneous appendages as well, with trichoblastic tumors originating from them. Furthermore, not all basal cell carcinomas show evident mucin production with artificial clefting towards the dermis, and mitotic activity can be conspicuous in some, especially smaller basal cell carcinomas. As both tumors fall into the category of blue basaloid dermal tumors with peripheral palisading and the mesenchymal component with formation of papillary mesenchymal bodies in some trichoblastomas is sparse, immunohistochemistry is often required, with trichoblastomas generally being diffusely positive for BerEp4 and basal cell carcinomas; while mostly showing positivity, the reaction is often patchy [3,11]. Studies have also underlined the role of cytokeratin 20 (CK20), which was not performed in our case, with most trichoblastomas showing a distinct Merkel cell component when stained with CK20, with basal cell carcinomas being sparse in most cases. As in our case, the tumor was located deep in the dermis, with no connection to the epidermis or appendages; the patient’s age and the tumor location, together with the pattern of BerEp4 expression, were considered sufficient to differentiate from basal cell carcinoma. Furthermore, around 40 cases of basomelanocytic tumor have been reported, which represent a mixed basal cell carcinoma and melanoma malignancy, in which both the epithelial and melanocytic components are malignant [12]. As seen in our case, the melanocyte component of the tumor comprised dendritic melanocytes, which had no morphological signs of malignant transformation.

The expression of BerEp4 also aided in the differential diagnosis of another rare pigmented and benign cutaneous appendageal tumor—Melanocytic matricoma [3]. To date, around 20 cases of melanocytic matricoma have been reported, with the tumor having a small predilection for the trunk, with most cases again in the head and neck area [13]. This tumor is a rare variant of pilomatricoma (Malhebre’s tumors), which has a prominent dendritic melanocyte component, similar to the one seen in our case, with some cases being poor in the diagnostic ghost/mummified cells for pilomatricomas. All pilomatricomas, however, are BerEp4-negative tumors, including the small subset of published melanocytic matricoma cases.

The last differential diagnosis of melanotrichoblastoma, due to its prominent pigment cell component, is that of melanoma [14,15]. While there is a melanocytic component within the tumor, both based on morphology and their immunophenotype characteristic, these are completely benign cells within the tumor, recapitulating the naturally occurring dendritic melanocytes of the hair follicles, which give out their melanin pigment to the hair shaft and color the hair. As such, melanotrichoblastomas not only represent a benign tumor, but can be viewed as the best differentiated variety of trichoblastoma as they completely reenact the structure of the hair follicle—epithelial, mesenchymal, and melanocytic components—with only the growth pattern differing from a normal hair follicle (Figure 4).

As trichoblastomas are often associated with sebaceous nevus, which in itself is not a true tumor but a hamartoma, due to the complete recapitulation of the hair follicle in melanotrichoblastoma, they can also be viewed as not only a mixed epithelial–mesenchymal-pigmented tumor, but probably as a type of hamartoma as well [9]. While melanotrichoblastomas are rare and no such conclusion or discussion has been raised in published cases so far, the authors hope that such a polemic debate can be achieved in future cases.

One further differential diagnosis was performed in our case, due to the high percentage of Ki-67 positive cells (high proliferative index) and the presence of degenerative type nuclear atypia in some centrally located cells, and that was with the malignant counterparts of trichoblastoma, trichoblastic carcinoma, and trichoblastic carcinosarcoma [16,17]. Both of these lesions probably originate from precursor trichoblastomas, with most of the handful of reported cases so far being either poorly circumscribed or ulcerated. In true trichoblastic carcinomas, only the epithelial component is malignant, with features of both cellular and nuclear atypia being moderate to pronounced on the background of high mitotic activity, with at least focal atypical mitosis and in some cases foci of necrosis [17]. In trichoblastic carcinosarcomas, of which fewer than a dozen cases have been published, both the epithelial and mesenchymal components have undergone malignant transformation [16]. In our case, despite mild atypia in the center of some nests and the centrally high proliferative index within the nests, only individual mitoses were noted throughout the tumor, with none of them being atypical, and the stromal component was completely benign.

As already mentioned, melanotrichoblastomas are exceedingly rare tumors, with the current case being one of less than ten published in the literature, to the best of the authors’ knowledge [10,18]. This, however, does not represent the true incidence of melanotrichoblastoma, but rather a rare and most likely underrecognized tumor [8]. The current WHO classification of skin tumors does not mention melanotrichoblastoma as a distinct entity, such as melanotic matricoma, nor does it mention it as a special subtype of trichoblastoma, rather stating that some trichoblastomas have melanin deposition, melanocyte colonization or hyperplasia, which probably further leads to these tumors being underreported and a high chance for some of them to be misdiagnosed with the entries as mentioned above.

Hopefully, future WHO skin tumor classifications will include melanotrichoblastoma as at least a specific subtype of trichoblastoma, if not a separate entry in itself, based on the accumulation of published cases.

## 4. Conclusions

Melanotrichoblastoma is a rare subtype of trichoblastoma, characterized by the presence of dendritic melanocytes within the tumor. These are rare and probably underrecognized tumors, with the currently depicted case being the seventh overall in the literature. Due to the striking pigmented nature of these tumors, both grossly and histologically, they require a broad differential diagnosis with other benign epithelial tumors as well as malignant epithelial and melanocytic ones.

## Figures and Tables

**Figure 1 reports-08-00130-f001:**
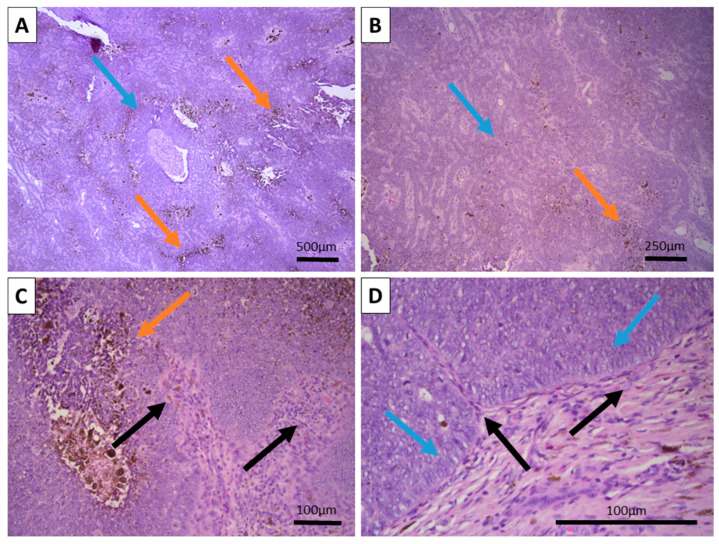
Histopathology of the tumor: (**A**) Large nodular growth pattern (blue arrow) and foci of extensive pigment deposition (orange arrows), H&E stain, original magnification 50×. (**B**) Reticular growth pattern (blue arrow) with foci of pigment deposition (orange arrow), H&E stain, original magnification 100×. (**C**) Wrapping hypercellular stroma with focal papillary mesenchymal bodies-like structures (black arrows) and intertumor pigment deposition (orange arrow), H&E stain, original magnification 200×. (**D**) Peripheral tumor cell palisading without clefting (blue arrows) towards the wrapping and intersecting hypercellular stroma (black arrows), H&E stain, original magnification 400×.

**Figure 2 reports-08-00130-f002:**
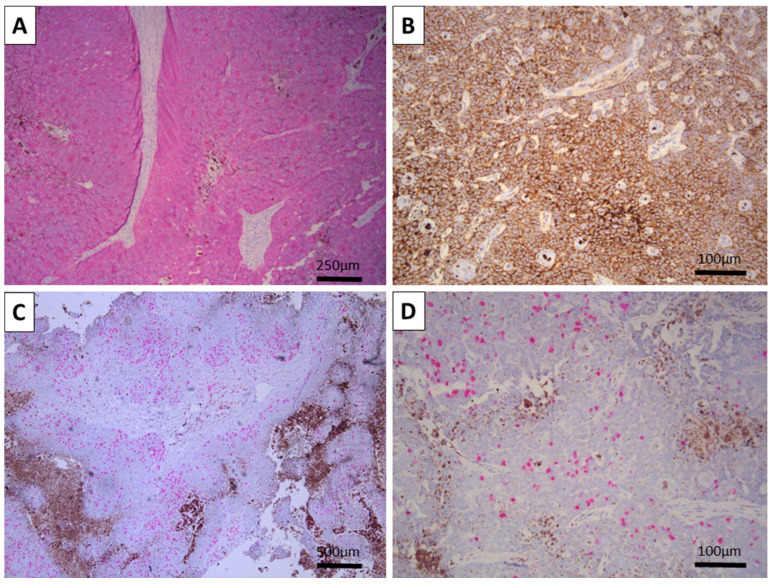
Immunohistochemistry for epithelial markers and proliferative index: (**A**) Diffuse positivity for CK AE1/AE3 within the tumor, note the stroma is negative, original magnification 100×. (**B**) Diffuse and strong positivity for BerEP4, original magnification 100×. (**C**,**D**) Foci of high (**C**) and intermediate to low (**D**) proliferative activity, original magnifications—100× (**C**) and 200× (**D**).

**Figure 3 reports-08-00130-f003:**
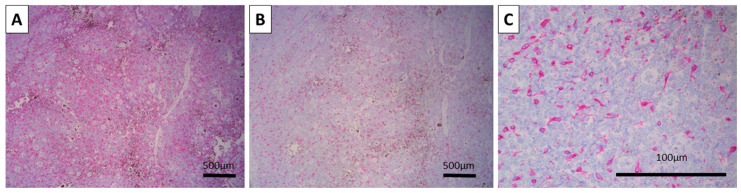
Immunohistochemistry for melanocytic markers: (**A**) Spindle cell melanocytes positive for HMB-45, original magnification 100×. (**B**,**C**) Spindle cell and dendritic morphology of the melanocytes within the tumor positive for Melan-A, original magnifications 100 (**B**) and 400× (**C**).

**Figure 4 reports-08-00130-f004:**
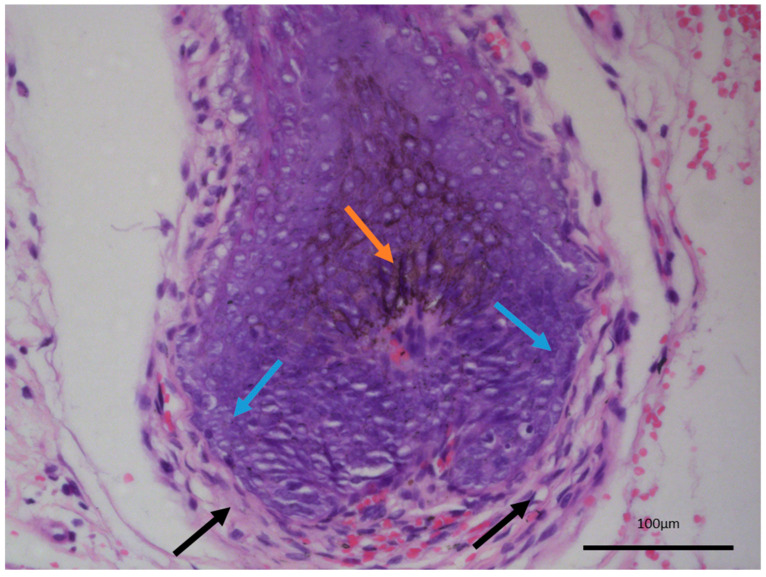
Normal hair follicle histology, internal control from adjacent non-involved tissues from the same patient. Dendritic melanocytes (orange arrow), palisading epithelial cells (blue arrows), and hypercellular wrapping stroma (black arrows), original magnification 200×.

## Data Availability

The original contributions presented in this study are included in the article. Further inquiries can be directed to the corresponding author.

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
