# Peer review of "Melanotrichoblastoma: A Histopathological Case Report of a Rare Pigmented Variant of Trichoblastoma"

_reports, 2025, doi:10.3390/reports8030130_

Round 1
Reviewer 1 Report
Comments and Suggestions for Authors
Dear Authors,
I was pleased to read this interesting Case Report. Melanotrichoblastoma is a rare tumor and it is under disscution if is different from pigmented trichoblastoma or not. That is precisely why I believe that presenting such cases could shed light on this subject.
Regarding the content of the article, I would have the following comments:
- as it is a rare condition, it is very important to present the clinical aspects of the tumor in such articles, in order to facilitate faster recognition of the diagnosis.
- i also saw that there no dermatoscopic image of the lesion?
- there are entire paragraphs referring to the same bibliographic reference (35-55)
- if there are few cases in the specialized literature, wouldn't it have been ideal to also include a table with their location or characteristics?
Author Response
Point-by-point response
Dear editors, staff and reviewers of Reports, thank you for considering our manuscript entitled “Melanotrichoblastoma: A Histopathological Case Report of a Rare Pigmented Variant of Trichoblastoma”; the following our point-by-point response to the comments made by the respected reviewers:
Comments of reviewer 1:
Dear Authors,
I was pleased to read this interesting Case Report. Melanotrichoblastoma is a rare tumor and it is under disscution if is different from pigmented trichoblastoma or not. That is precisely why I believe that presenting such cases could shed light on this subject.
- Dear reviewer, thank you for this kind comment regarding our manuscript.
Regarding the content of the article, I would have the following comments:
as it is a rare condition, it is very important to present the clinical aspects of the tumor in such articles, in order to facilitate faster recognition of the diagnosis.
- Dear reviewer, thank you for this comment. We will expand the section of patient symptoms in the revised version, although admittedly they are broadly unspecific and there is nothing much to add in this regard
i also saw that there no dermatoscopic image of the lesion?
- Sadly, we have no dermatoscopic image for the case. We hope this drawback will not be considered significant enough to hinder the chances of the manuscript for publication. As we are aware of this drawback, which is why we specifically focused on the histological aspents, hence A Histopathological Case Report is part of the manuscript title.
there are entire paragraphs referring to the same bibliographic reference (35-55)
- Thank you for this comment. These are the only seven manuscript in the published literature that have identified and discuss the nosological unit at hand, hence the multiple repetition of the same references.
if there are few cases in the specialized literature, wouldn't it have been ideal to also include a table with their location or characteristics
- Thank you for this comment. Yes, this highly ideal, however the sixth published case has already produced such a table and since its publication there has only been one more case published. IF we were to produce such a table, then we would already repeat what has already been summarized by Ocanha-Xavier and Xavier-Júnior in 2023 (Ocanha-Xavier, J.P.; Xavier-Júnior, J.C.C. Melanotrichoblastoma: Sixth Case Report in the Literature. An. Bras. Dermatol. 2023, 98, 871–874, doi:10.1016/J.ABD.2022.07.011,)
We hope the changes are sufficient and satisfy the requests of the respected reviewer and that the manuscript can be forwarded for publication.
If any further changes are needed for the manuscript, please let us know and we will implement them ASAP.
With respect,
George S. Stoyanov, MD, PhD
Corresponding author
21.07.2025
Shumen, Bulgaria
Reviewer 2 Report
Comments and Suggestions for Authors
Dear authors,
thank you for your valuable case, which expands the anatomical spectrum of an extremely rare adnexal tumor with prominent melanocytic colonization.
I suggest some additions:
- Specify the duration of follow-up (months) and the date of the last evaluation
- Provide a structured and in-depth discussion of the possible differential diagnoses (pigmented BCC, melanocytic matricoma, melanoma, trichoblastic carcinoma) using architectural + stromal contrasts + IHC (please see 10.3390/jpm14050518)
- Provide operational criteria or diagnostically flowchart to distinguish melanotrichoblastoma from pigmented trichoblastoma and collision lesions
- Add scale bars, low-power overview, uniform legends, and correct spelling ("reticular")
- Correct typos (Bartolini, appendix, criteria), simplify the abstract, standardize bibliographic references/DOIs.
Author Response
Point-by-point response
Dear editors, staff and reviewers of Reports, thank you for considering our manuscript entitled “Melanotrichoblastoma: A Histopathological Case Report of a Rare Pigmented Variant of Trichoblastoma”; the following our point-by-point response to the comments made by the respected reviewers:
Comments of reviewer 2:
Dear authors,
thank you for your valuable case, which expands the anatomical spectrum of an extremely rare adnexal tumor with prominent melanocytic colonization.
- Dear reviewer, thank you for this kind comment towards our manuscript.
I suggest some additions:
Specify the duration of follow-up (months) and the date of the last evaluation
- Thank you for this suggestion, follow-up period has been added.
Provide a structured and in-depth discussion of the possible differential diagnoses (pigmented BCC, melanocytic matricoma, melanoma, trichoblastic carcinoma) using architectural + stromal contrasts + IHC (please see 10.3390/jpm14050518)
- Dear reviewer thank you for this comment. The discussion section already structured as such and included the following entries: BCC and its variant, including basomelanocytic tumor – a collision between BCC and melanoma; melanotic matricoma; melanoma and trichoblastic carcinoma/carcinosarcoma. The differentials are discussed both in the context of cell morphology and that of immunohistochemical expression. In this regard we do not understand the extent or requirement for discussion expansion, for which we must apologize.
Provide operational criteria or diagnostically flowchart to distinguish melanotrichoblastoma from pigmented trichoblastoma and collision lesions
- As there are very few cases published, and at the time being the diagnosis is that of exclusion, we do not believe that sufficient evidence has been published in the literature that will allow us to produce an objective flowchart for the diagnosis of melanotrichoblastoma, which will not sway colleagues into making over diagnosis of this rare and benign entry
Add scale bars, low-power overview, uniform legends, and correct spelling ("reticular")
- Thank you for this suggestion. We have added scale bars and modified the figure legends
Correct typos (Bartolini, appendix, criteria), simplify the abstract, standardize bibliographic references/DOIs.
- Thank you for noticing these errors made by us, they have been revised in new version of the manuscript. References without relevant DOI do not have one provided by the original publisher.
We hope the changes are sufficient and satisfy the requests of the respected reviewer and that the manuscript can be forwarded for publication.
If any further changes are needed for the manuscript, please let us know and we will implement them ASAP.
With respect,
George S. Stoyanov, MD, PhD
Corresponding author
21.07.2025
Shumen, Bulgaria